# Combining phylogeny and coevolution improves the inference of interaction partners among paralogous proteins

Carlos A. Gandarilla-Pérez[1,2], Sergio Pinilla[2,3¤], Anne-Florence Bitbol [4,5]*, Martin Weigt [2]*

**1** Facultad de Física, Universidad de la Habana, San Lázaro y L, Vedado, Habana, Cuba, **2** Sorbonne Université, CNRS, Institut de Biologie Paris-Seine, Laboratoire de Biologie Computationnelle et Quantitative (LCQB, UMR 7238), Paris, France, **3** Sorbonne Université, CNRS, Institut de Biologie Paris-Seine, Laboratoire Jean Perrin (UMR 8237), Paris, France, **4** Institute of Bioengineering, School of Life Sciences, École Polytechnique Fédérale de Lausanne (EPFL), Lausanne, Switzerland, **5** SIB Swiss Institute of Bioinformatics, Lausanne, Switzerland

¤ Current address: Heuritech, Paris, France
* anne-florence.bitbol@epfl.ch (A-FB); martin.weigt@sorbonne-universite.fr (MW)

**Data Availability Statement:** Matlab implementations of the DCA-based IPA and the MI-based IPA on our standard HK-RR dataset are freely available at https://doi.org/10.5281/zenodo.

## Abstract

Predicting protein-protein interactions from sequences is an important goal of computational biology. Various sources of information can be used to this end. Starting from the sequences of two interacting protein families, one can use phylogeny or residue coevolution to infer which paralogs are specific interaction partners within each species. We show that these two signals can be combined to improve the performance of the inference of interaction partners among paralogs. For this, we first align the sequence-similarity graphs of the two families through simulated annealing, yielding a robust partial pairing. We next use this partial pairing to seed a coevolution-based iterative pairing algorithm. This combined method improves performance over either separate method. The improvement obtained is striking in the difficult cases where the average number of paralogs per species is large or where the total number of sequences is modest.

## Author summary

When two protein families interact, their sequences feature statistical dependencies. First, interacting proteins tend to share a common evolutionary history. Second, maintaining structure and interactions through the course of evolution yields coevolution, detectable via correlations in the amino-acid usage at contacting sites. Both signals can be used to computationally predict which proteins are specific interaction partners among the paralogs of two interacting protein families, starting just from their sequences. We show that combining them improves the performance of interaction partner inference, especially when the average number of potential partners is large and when the total data set size is modest. The resulting paired multiple-sequence alignments might be used as input to

1421861 and https://doi.org/10.5281/zenodo.1421781, respectively. Julia implementations of the GA-IPA (as well as of the IPA), both DCA-based and MI-based, are freely available at https://doi.org/10.5281/zenodo.7731108.

**Funding:** CAGP and MW acknowledge funding by the EU H2020 Research and Innovation Programme MSCA-RISE-2016 (grant agreement No. 734439 InferNet). AFB acknowledges funding by the European Research Council (ERC) under the EU H2020 Research and Innovation Programme (grant agreement No. 851173). AFB and MW thank the Institut de Biologie Paris-Seine (IBPS) at Sorbonne Université for funding via a Collaborative Grant (Action Incitative). This work was performed in part at the Aspen Center for Physics, which is supported by National Science Foundation grant PHY-1607611. The funders had no role in study design, data collection and analysis, decision to publish, or preparation of the manuscript.

**Competing interests:** The authors have declared that no competing interests exist.

machine-learning algorithms to improve protein-complex structure prediction, as well as to understand interaction specificity in signaling pathways.

## Introduction

Sequence-driven modeling and prediction techniques for proteins have recently seen great advances, thanks to the combination of the rapidly growing amount of available protein-sequence data, with powerful statistical and machine learning techniques. Recently, AlphaFold brought a major advance in protein-structure prediction for monomeric proteins, provided that a sufficiently large number of homologous proteins can be found [1]. Indeed, AlphaFold starts from multiple-sequence alignments (MSAs) of homologs. Extensions to multimers and protein-protein interactions have been proposed, and they also start from MSAs of homologs of the proteins involved [2–4]. While these advances are impressive, many protein complexes remain unsolved by current computational means. A possible direction to improve them is to produce better co-alignments of interacting proteins (co-MSAs), where each row contains the concatenation of two interacting proteins. Indeed, co-MSAs allow to exploit correlations between interaction partners, which convey important information about binding specificity [5]. Beyond the perspective of improving quaternary protein-structure prediction, being able to accurately pair interacting paralogs is important to unveiling signaling networks and to understanding interaction specificity. Indeed, homologous signaling pathways in a given organism employ homologous mechanisms, but crosstalk between pathways may be unwanted. This is the case for instance in two-component systems in bacteria [6, 7], or calcium signaling in plants [8, 9].

Pairing interacting paralogs and obtaining co-MSAs is difficult because many protein families contain several paralogous proteins encoded within the same genome. Out of the almost 20,000 protein domain families in the Pfam database [10], now integrated into InterPro [11], more than 1300 have, on average, at least 10 paralogs per species, and about 400 more than 30 paralogs. These strongly amplified protein families include, beyond repeat domains which are not the most relevant here, many specifically interacting protein families like receptors, transporters, kinases etc., used, e.g., in signal transduction.

Therefore, even if we know that two protein families A and B interact, it can still be difficult to determine which particular paralog in family A interacts with which one in family B. In some cases, the problem can be solved using genomic co-localization of the protein-coding genes, as in operons in bacteria, and we will use such cases as benchmark cases for our algorithm. If this does not apply to the proteins studied, e.g. because they are not in the same operon, or because they are eukaryotic, the paralog-pairing problem becomes substantially more challenging. In practice, large-scale coevolution-based studies of inter-protein structural contacts [12] and protein-protein interactions [13, 14], as well as recent deep learning-based predictions of quaternary structures [2, 4], rely on co-MSAs constructed using genomic co-localization [12, 15] when possible, and orthology, determined by reciprocal closest matching sequences [2–4, 13, 14]. However, restricting to orthologs reduces co-MSA depth compared to using all paired paralogs.

Here, we propose to combine two important evolutionary signals, namely phylogeny and residue co-evolution, in order to improve paralog pairing. First, phylogeny, or in practice sequence similarity, can be helpful because when two proteins A and B interact in one species, and possess close homologs A', B' in a second species, then A' and B' are likely to also interact. More generally, the phylogenies of interacting protein families are expected to be similar. This

idea has been used to discriminate interacting from non-interacting families starting from protein orthologs in the MirrorTree family of approaches [16–18], some of which address the paralog pairing problem [19–28]. The use of orthology for co-MSA construction [2, 4, 13, 14] also relies on this idea. One of us proposed to employ the similarity of phylogenies through neighbor-graph alignment for paralog pairing, but performance remained limited [26]. Second, interacting proteins coevolve (see [5, 29, 30] for reviews), which can be employed to predict interaction partners, as first demonstrated in a Bayesian network approach in [31]. Coevolutionary modeling approaches like Direct Coupling Analysis (DCA) [15, 32, 33] or Gremlin [12, 13] have been used successfully to infer inter-protein structural contacts from co-MSAs. More recently, some of us showed that iterative algorithms based on DCA allow paralog pairing [34, 35]. However, while these methods can function even without an initial seed co-MSA, their performance remains limited for high paralog numbers as well as for small datasets [34]. Interestingly, these coevolution-based methods already benefit from phylogenetic correlations [36, 37], and mutual information, which uses all forms of statistical dependences, performs slightly better than DCA for the pairing task [38]. These results show the potential of combining phylogenetic and coevolutionary signals. Coevolution-based methods are unlikely to exploit sequence similarity in an optimal way, because they are not designed for that.

Here, we show that explicitly combining these two signals substantially improves the pairing of paralogs between interacting protein families. We first use phylogenetic signal, by aligning orthology or sequence-similarity graphs, to produce an accurate co-MSA spanning a subset of the data. For this, we employ a stochastic algorithm based on simulated annealing to solve this graph-alignment (GA) problem. We compare the results of GA to those of a MirrorTree-based approach. We next use the partial co-MSA obtained from similarity graph alignment as a seed for the DCA-based iterative pairing algorithm (IPA). Our method, called GA-IPA as it combines GA and IPA, is illustrated in Fig 1. We obtain high-quality co-MSAs, outperforming those obtained by methods based on sequence similarity (phylogeny) or on coevolution alone. We use bacterial two-component systems as a specific example, and we also apply our approach to other protein families, thus showing its robustness and broader applicability.

## Results and discussion

### Goal

We start from two MSAs comprising the sequences of two interacting protein families A and B (see Fig 1A). We assume that each species comprises the same number of paralogs of family A and of family B. While this is a simplification with respect to typical protein families, we make this choice for two reasons: (i) our benchmark data sets, generated using genome proximity, possess this property, and (ii) generalizing the method to the unbalanced case is rather straightforward, but the formulation is more involved. We therefore focus on the simpler case that is directly testable with our benchmark data.

Using these MSAs, and only these MSAs, we aim at constructing a bijective *paralog pairing* (or matching) which assigns to each sequence in family A one putative interaction partner in family B from the same species. Potential inter-species PPI are thus discarded by our algorithm, but could be detected afterwards with techniques similar to IPA.

In order to construct this pairing, we will use successively two distinct methods. The first one exploits phylogenetic relationships between interaction partners, via sequence similarity (i.e., it aims at identifying interologs). The second, and more involved one, relies on the inter-family coevolutionary signal as detectable by DCA (or mutual information). Both methods

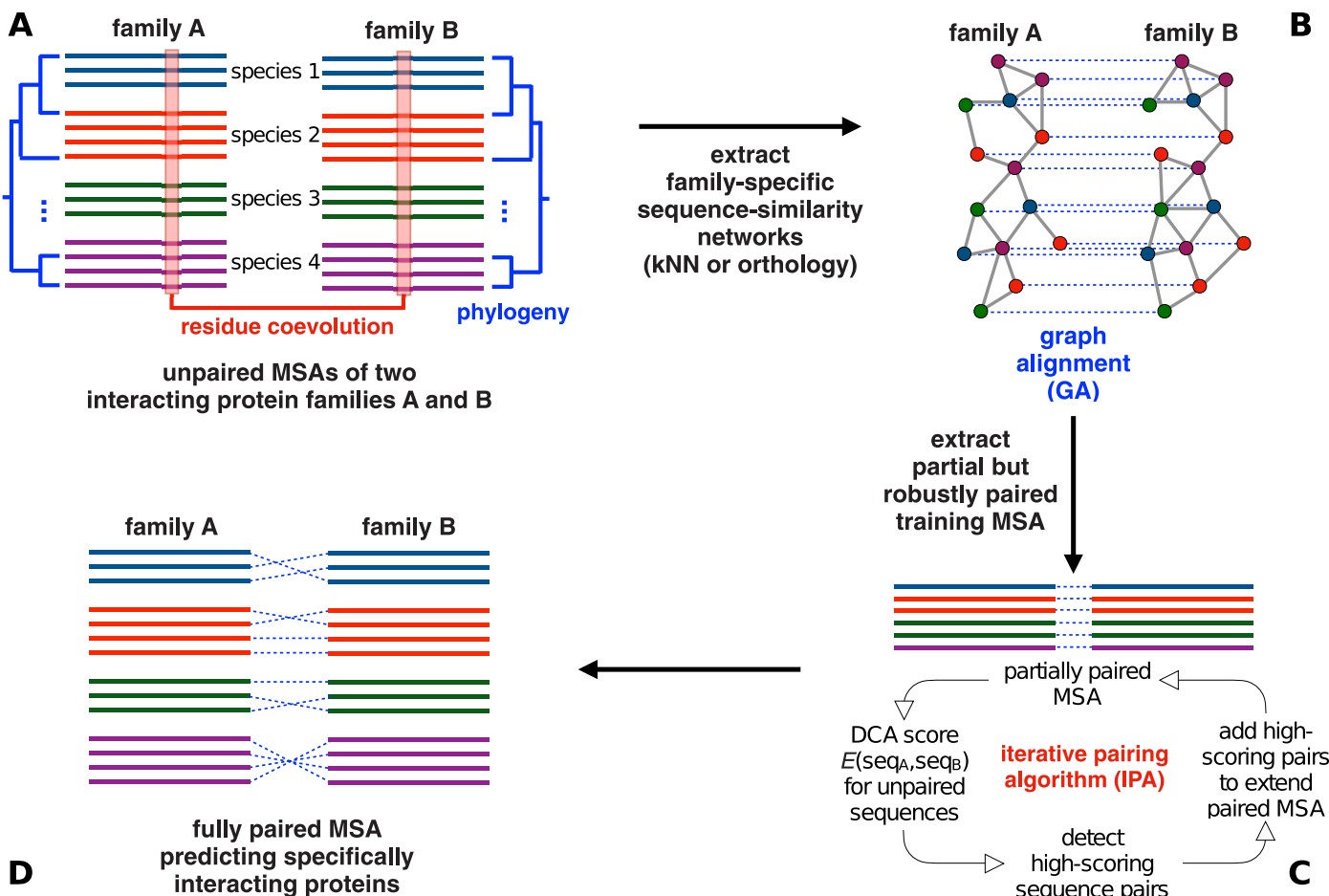

**Fig 1. GA-IPA approach for paralog pairing between two interacting protein families.** We show schematics to illustrate the key points of our approach (these schematics are conceptual and do not represent actual phylogenetic trees or similarity networks). (A) One starts from the separate MSAs of two interacting protein families A and B. Each species present in these MSAs may contain multiple paralogs in each family. Our goal is to infer which paralog in family A interacts with which paralog in family B. As indicated, two types of information will be used: phylogeny and residue coevolution. (B) We first construct a sequence-similarity network, specifically a k nearest-neighbor (kNN) or an orthology network, for each of the two families. These two networks are aligned to find a pairing of the sequences that maximises the similarity of the two networks, while only allowing pairs within the same species. Repeated runs of a stochastic graph-alignment (GA) algorithm based on simulated annealing allow to identify robust pairs, which are consistently paired across GA runs. (C) This partial but robustly paired MSA is used as an input to the iterative pairing algorithm (IPA) based on residue coevolution as detected by DCA. IPA iteratively extends the paired MSA until all sequences are paired. (D) The output full co-MSA is our prediction for the interacting protein pairs between families A and B.

lead to computationally hard optimization problems (by contrast, e.g., to standard binary matching), which we approximately solve by heuristic techniques.

## Aligning sequence-similarity networks to identify a subset of robust paralog pairings

In the first step, we aim at using phylogenetic relationships to identify potential interaction partners. Specifically, we exploit phylogenetic relations by constructing *sequence-similarity networks* for each of the two families A and B, and by aligning them together (see Materials and methods for details). We use the Hamming distance, which simply counts the amino-acid mismatches between two aligned sequences. It could be replaced by more sophisticated distance measures, including, e.g., amino-acid similarities or a position-specific weighting based on conservation in MSA columns. However, since our sequences are already well aligned

(using profile Hidden Markov Models [39]), we expect that this would not substantially change our results. Sequence-similarity networks are weighted undirected graphs, where each node corresponds to a sequence. We consider two types of similarity networks. First, in the *kNN network*, each node is connected to its $k$ nearest neighbors, where $k$ is a parameter that will be varied in our study. Second, in the *orthology network*, each protein is connected to its orthologs, identified using best reciprocal hits, in other species. These networks are weighted in a way that gives stronger importance (i.e. larger weight) to edges between similar sequences than to edges between distant sequences.

Because of the similarities in the phylogenies of interacting proteins (see above), edges between the two similarity networks associated to each of the two families A and B tend to overlap, if the nodes representing partners are paired. We therefore map the search for a paralog pairing to a *graph-alignment* problem, where the vertices of the two graphs are paired to maximize the number of overlapping edges (see Materials and methods). To solve this difficult problem, we introduce a heuristic stochastic local search algorithm based on *simulated annealing* (cf. *Materials and methods*).

Fig 2 shows results for a benchmark set of $M = 5064$ bacterial two-component systems from $N = 459$ species, describing the interaction between histidine kinases (HK) and response regulators (RR), cf. *Materials and methods* for details of the dataset and the corresponding similarity networks. Similar results are obtained for other protein-family pairs, as we report in the *Supporting information*.

For the orthology network, we find on average about 2/3 of correct pairings (true positives, TP), and 1/3 of incorrect pairings (false positives, FP), with higher TP fractions corresponding in general to lower GA cost, as is shown in the *Supplementary information*. This is much better than a random within-species pairing, which would have an average TP fraction of only $N/M \simeq 9\%$. For the kNN network, we find a strong dependence on $k$. Indeed, for small $k$, only few edges can be aligned, since they connect different species, and the similarity networks contain little information about the correct paralog pairing. For larger values of $k$, the results slightly outperform those of the orthology network, with an average TP fraction of about 70% (see Fig 2). However, the increase of accuracy with increasing $k$ comes with the higher computational cost of aligning denser similarity networks.

Fig 2 shows that our GA results reach, on this HK-RR dataset, a lower performance than the IPA starting from random matchings, i.e. without any training set of known paired sequences. Instead of sequence similarity networks, the IPA takes full sequences as input, and benefits from phylogenetic correlations, as we have shown recently [36, 37]. The IPA reaches on average 84% of TPs, outperforming even the best of the numerous GA runs done for Fig 2.

An interesting observation can be made when comparing the pairings $\pi$ resulting from multiple runs of our stochastic GA algorithm. As is reported in Fig 3A, about 1300 HK sequences are paired across all runs with exactly the same RR protein. In this robust subset of paired sequences, a very high fraction of 99% of TPs is reached. Thus, almost all FPs appear among non-robust pairs, which differ from one GA run to the other. Fig 3B further shows that, while the number of robust pairings depends on the particular similarity network used, the TP fractions within the robust subset are always very high. Note that, in terms of the size and the quality of the robust subset, kNN networks outperform the orthology network even at moderate $k$ for which the overall accuracy of the orthology networks for GA is still superior.

We also compare the results of our GA method to those of the related MirrorTree family of approaches [16–18], which aim to predict protein-protein interactions by exploiting sequence similarity (and thus phylogeny), see Materials and methods. Indeed, some MirrorTree-based methods have tackled the paralog pairing problem [19–28]. To make a comparison, we implemented such a method (see Materials and methods for details). Table A in S1 Text shows that

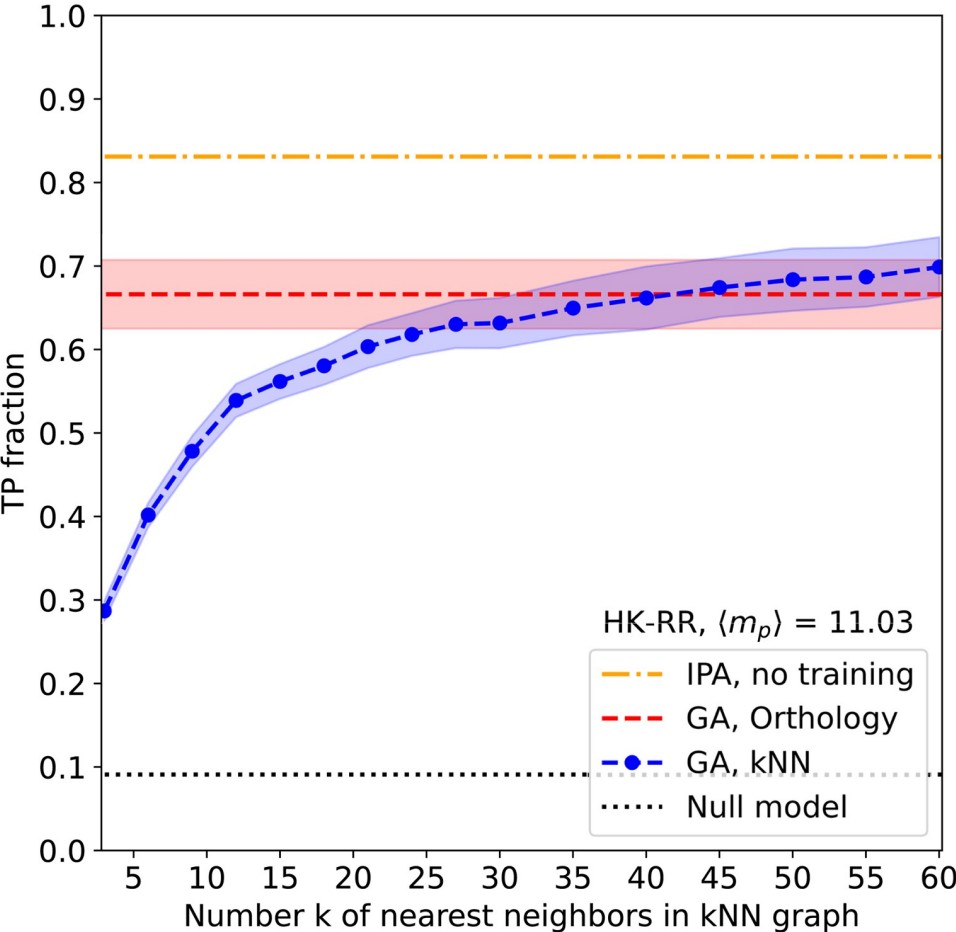

**Fig 2. Performance of graph alignment.** The mean fraction of true-positive pairings (TP fraction) is shown as a function of the number $k$ of nearest neighbors in the kNN graph for 100 GA realizations (blue). Performance for the orthology graph is also shown (red)—but note that it does not depend on $k$. Error bars (shaded regions) correspond to one standard deviation. The mean TP value of the IPA starting without any training set of known paired sequences is shown for comparison (yellow). It was obtained using $N_{increment} = 6$ and by averaging over 50 replicates that differ in their initial random pairings, cf. *Materials and methods*. The dotted black line shows the average TP fraction obtained for random HK-RR pairings within species (null model).

the performance of GA and of MirrorTree is similar for several datasets, but we find a substantially better TP fraction using GA than MirrorTree for the standard HK-RR dataset with $\langle m_p \rangle = 11.03$. While GA and MirrorTree share conceptual similarities, one key difference is that GA mainly focuses on close neighbor sequences, by restricting to the $k$ nearest neighbors, or to reciprocal closest matches, and by giving exponentially decreasing weights to distant pairs (see Materials and methods). Conversely, MirrorTree relies on all pairs, including many informatively distant ones, which may become problematic for large datasets and with large numbers of paralogs per species. This may explain the difference observed for the HK-RR dataset. In the MirrorTree approach, one can exploit the stochasticity of the optimization to define robust pairs, exactly as for GA. Table B in S1 Text shows that MirrorTree yields similar numbers of robust pairs as GA but with a lower TP fraction.

To summarize, GA of the sequence-similarity networks of the two MSAs of interest allows to identify a robust subset of paired sequences and to construct a robust partial co-MSA with

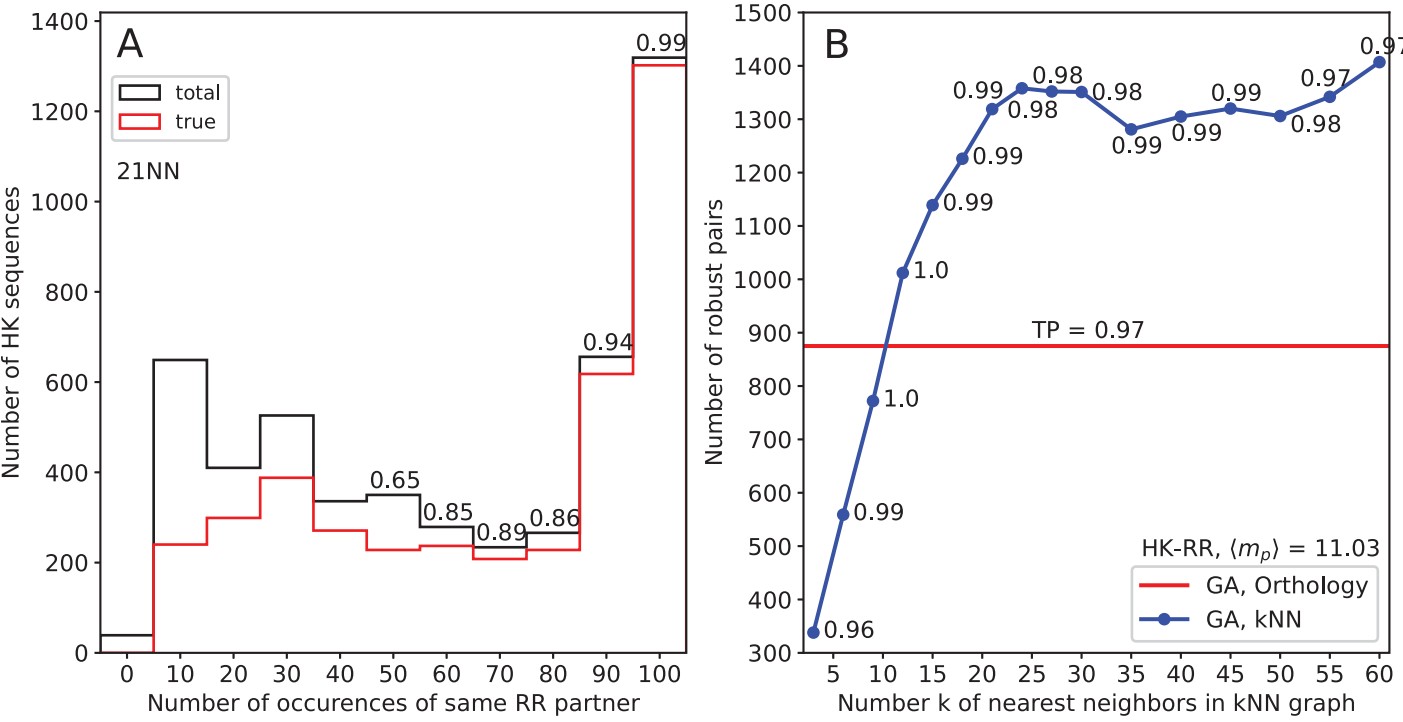

**Fig 3. Robustness of GA.** (A) Robustness histograms for the 21-NN graph. We perform 100 GA runs and count how many times a HK is paired to the same RR. The horizontal axis gives the number of times a given pair appears, and the vertical axis is the number of pairs appearing that many times across replicates. Black bars are the total number of pairs, and red bars are the number of TPs. The TP ratios are indicated on top of the bars. (B) Number of robust pairs (occurring in all 100 GA runs) obtained by GA for each similarity network. The fraction of correctly matched pairs in this robust subset is indicated in each case.

high accuracy. Furthermore, the accuracy of this robust partial co-MSA is higher when using GA than when using MirrorTree. Next, we employ this partial co-MSA as a starting point for the IPA, and use the IPA to extend it to a full co-MSA.

## Using a robust partial co-MSA to seed the coevolution-based iterative pairing algorithm (IPA) yields accurate co-MSAs

The IPA was introduced in [34], and the idea of this method is shown in Fig 1C and detailed in *Materials and methods*. Briefly, this algorithm starts from a seed co-MSA and employs a coevolution-based DCA model [15, 32, 33] built on this seed co-MSA to score all possible within-species A-B pairs and propose a one-to-one matching of sequences. Proposed pairs with top confidence scores are added to the co-MSA to improve the DCA model and the predicted pairings, and this procedure is iterated, growing the co-MSA by $N_{\text{increment}}$ concatenated sequences at each iteration, until a full co-MSA is obtained. The IPA thus constructs a pairing that approximately maximizes DCA coevolutionary signal.

The IPA can also be run without a seed co-MSA [34], as is done e.g. in Fig 2. This serves as a baseline comparison for our combined GA-IPA approach. In this case, a random pairing within each species is used to infer the first DCA model. Since this random matching has on average one correct pair per species, we expect some coevolutionary signal to emerge if the average number of paralogs per species $\langle m_p \rangle = M/N$ is not too large, but the total dataset depth $M$ is large [40, 41]. The resulting DCA model is used to find the $N_{\text{increment}}$ highest-scoring

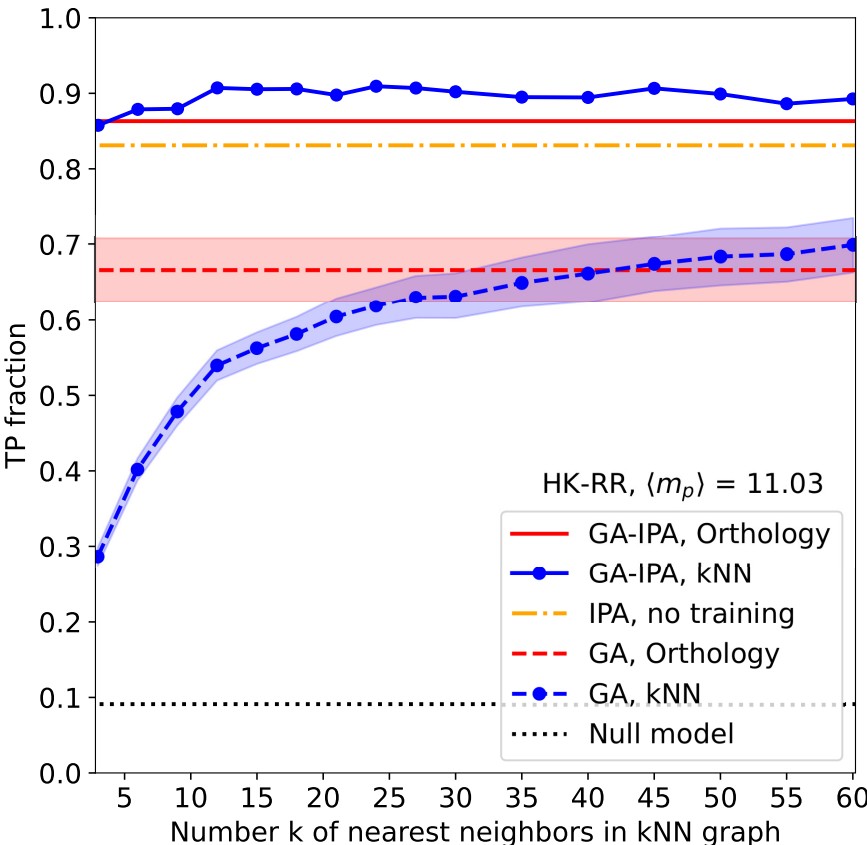

**Fig 4. GA-IPA outperforms both GA and IPA.** The mean fraction of true-positive pairings (TP ratio) is shown as a function of the number of nearest neighbors $k$ in the kNN graph. We combine GA and IPA in our GA-IPA method: we use the robust pairs obtained by GA as a seed co-MSA for IPA. The results of GA and of IPA without seed co-MSA from Fig 2 are shown for comparison. For IPA, we use $N_{increment} = 6$, both without (IPA) and with (GA-IPA) seed co-MSA.

pairs, which in the second step are used to replace the randomly matched MSA. Iterations are then continued as described in the previous paragraph. The fact that signal adds more constructively than noise, and the fact that two interacting pairs from different species tend to be more similar to one another than to non-interacting pairs, both help to bootstrap IPA toward high performance starting from random pairings (see Fig 2). It has however been shown that the performance of IPA strongly increases when starting from a seed co-MSA, in particular for data sets with large numbers of paralogs or small depth, where starting from random pairings does not yield good performances [34].

This last observation is important in our context. Indeed, using GA without any seed co-MSA, we have constructed a robust partial co-MSA, which we found to be highly accurate. This partial co-MSA from GA can now be used as a seed co-MSA for IPA. We call this new method GA-IPA, since it combines GA and IPA.

Fig 4 shows the performance of GA-IPA for both the orthology graph and the kNN graphs with various $k$. We find that the results of our combined GA-IPA method are substantially better than those of each separate algorithm. GA-IPA even outperforms the already quite accurate IPA results (up to 90% TP fraction in GA-IPA vs. 84% in IPA). Second, results are very robust across different similarity networks. Even for the small value $k = 3$, when GA alone has a low

TP rate and produces a small robust partial co-MSA, we obtain very good results: IPA is able to benefit even from pretty small seed co-MSAs.

We applied GA-IPA to other datasets corresponding to two other pairs of interacting protein families with different biological functions (ABC transporters and enzymes), as well as to two pairs of protein families with no known interaction but encoded in close proximity in prokaryotic genomes (see Materials and methods). Results are shown in Fig C in S1 Text. We find a very good performance of GA-IPA in all cases, showing its robustness, but the gain of performance compared to IPA alone is very limited. Indeed, the limited numbers of paralogs per species in these datasets make the IPA already very efficient and robust without any seed co-MSA. Note that the successful performance of IPA on pairs of protein families with no known interaction shows that it is already able to exploit phylogenetic signal [36, 37]. With GA, this signal is exploited more explicitly.

As mentioned above, another way of taking into account sequence similarity (and thus phylogeny) is to use MirrorTree approaches. We compare the results of our methods to variants using MirrorTree scores in Table C in S1 Text. We consider two types of such variants (see Materials and methods). First, we use MirrorTree to build the starting robust partial co-MSA (see above and Table B in S1 Text), and we run the DCA-based IPA starting from it. Second, we use MirrorTree scores instead of DCA scores within the IPA, either without a training set, or starting from robust partial co-MSAs from either GA or MirrorTree. We find that these MirrorTree-based variants perform slightly less well than the DCA-based IPA for all datasets, and significantly less well for the HK-RR dataset.

## GA-IPA yields accurate co-MSAs for data sets with few sequences or high paralog multiplicities

In Fig 4 and Fig C in S1 Text, the benefit of using GA-IPA instead of the standard IPA without seed co-MSA remains limited. However, these are cases where the IPA already reaches a high accuracy without any seed co-MSA. We now address two hard cases where the IPA without seed co-MSA has poor performance.

The first difficult case we consider involves high average multiplicities $\langle m_p \rangle = M/N \gg 1$ of paralogs per species, making the pairing task very hard. To investigate this case, we constructed a data set by selecting species with large paralog numbers in our data set of two-component system protein sequences, cf. *Materials and methods*. The results for $\langle m_p \rangle = 29.2$, i.e. for almost three times more paralogs per species than in the previous dataset, are shown in Fig 5A. In this case, a random matching has only 3.4% TPs. The IPA without seed alignment reaches 16% TP rate, which is better than random, but not sufficient for practical applications. GA alone already performs better in this case: kNN similarity networks with large enough $k$ reach about 40% TP rate. An interesting result is obtained when combining the two: GA-IPA reaches almost the same TP rate (80–90%) as with the data set considered in Fig 4, which however had only an average of 11 paralogs per species. Besides, for this dataset, our approach performs substantially better than MirrorTree-based variants, see Table C in S1 Text. We conclude that GA-IPA is very robust to high paralog multiplicities. This provides a major improvement over previous approaches, and extends the applicability of paralog pairing to highly amplified protein families, cf. the *Introduction*.

The second difficult case we consider is the case of small MSAs (i.e. MSAs with small $M$). To analyse this case, we randomly subsampled the species in the full HK-RR data set (see Materials and methods) to obtain smaller data sets, some being as small as $M = 100$ sequences. As can be seen in Fig 5B, the IPA without seed co-MSA has a strong $M$ dependence, and several thousands of sequences are needed in each MSA to reach high TP rates above, e.g., 70–

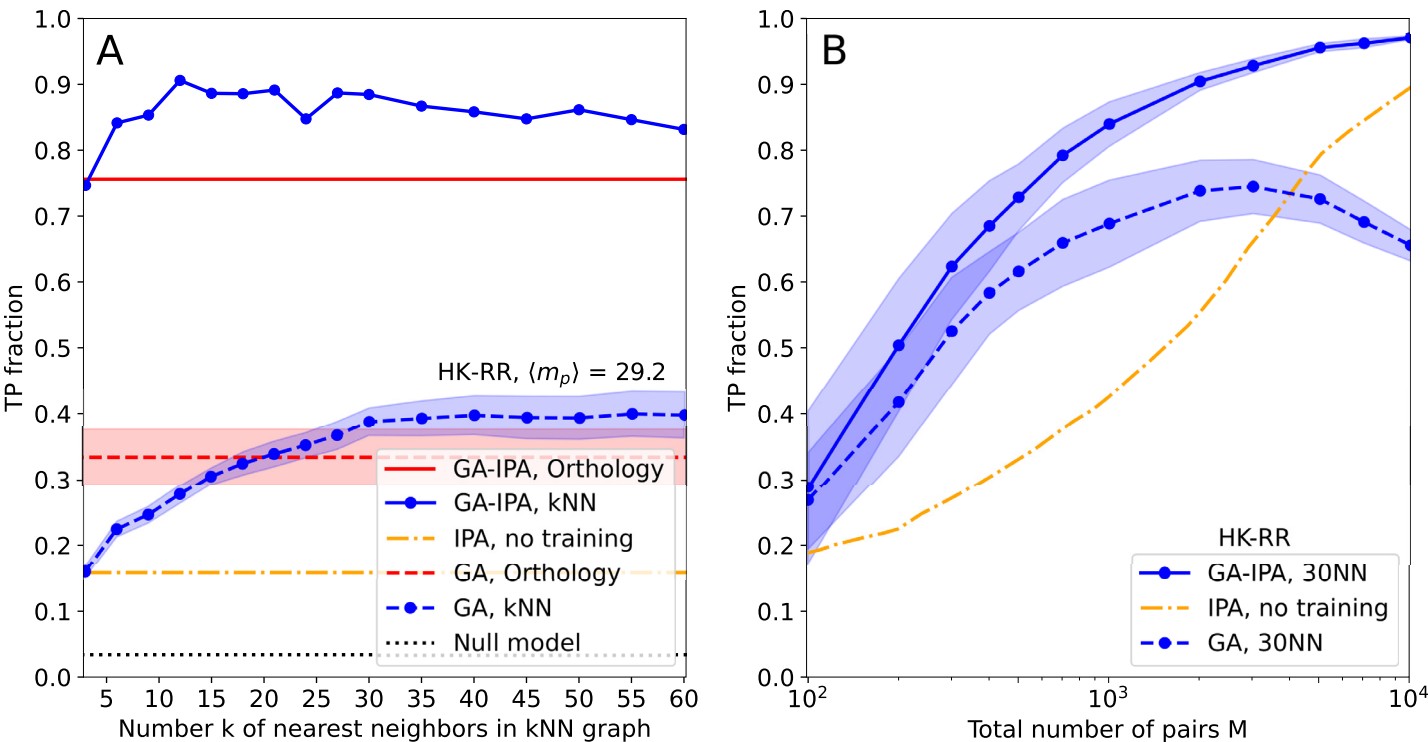

**Fig 5. Robust performance of GA-IPA in hard cases of paralog pairing.** (A) Same as Fig 4 but on a dataset having on average 29.2 paralogs per species, compared to 11.03 in Fig 4. While the performance of GA is substantially reduced compared to Fig 4, and that of IPA is even more reduced, GA-IPA achieves much larger TP fractions than GA and IPA. (B) Results of GA, IPA and GA-IPA for smaller datasets obtained by species subsampling from the full HK-RR data set, with 11.1 paralogs per species on average. We observe that GA-IPA needs almost one order of magnitude less sequences than IPA to reach comparable TP fractions.

80%. For small $M$, GA performs a bit better, and its TP rate increases faster, outperforming the IPA by a factor larger than two for $M$ on the order of a few hundreds. However, GA yields smaller TP rates for larger $M$ (at constant $k$ for the kNN graph), and GA performs substantially less well than the IPA for $M$ on the order of a few thousands. Indeed, when $M$ increases, more and more edges of the similarity network link distinct species in the two families, making the networks difficult to align. Crucially, we find that the combined GA-IPA always performs best, thus getting the best of both worlds. For very small data sets, GA-IPA does not lead to substantial accuracy gains over GA alone, because DCA needs sufficient data for accurate inference. But as soon as $M$ is about a few hundreds, GA-IPA outperforms GA, and it does not suffer from the decay of GA performance at large $M$, as the extracted robust partial pairings of GA are sufficient to seed IPA, which itself performs better for larger $M$ even without seed co-MSA. To reach TP rates of 70% or 80%, GA-IPA only needs about one thousand of sequences, compared to several thousands for the IPA without seed co-MSA.

In Fig D in S1 Text, we perform the same analysis as in Fig 5 with the mutual information (MI)-based IPA introduced in [38] instead of the DCA-based IPA [34]. It shows that our results hold for the MI-based IPA as well as for the DCA-based IPA. In both cases, using the robust partial seed co-MSA from GA yields a substantial increase of performance. Without a training set, the MI-based IPA requires slightly less data in total to achieve good performance than the DCA-based one [38]. For GA-IPA, this difference becomes smaller, but the MI-based IPA still very slightly outperforms the DCA-based one (see Fig D in S1 Text).

**Computational time.**    GA is usually less computationally costly than IPA (see Fig E in S1 Text). Furthermore, using a training set reduces the number of required IPA iterations compared to starting from no training set. However, in order to get a robust seed co-MSA, we employ multiple runs of GA, which increases the computational cost of the GA step, but is amenable to naive parallelism. Overall, our results show that using GA to provide a training set for the IPA yields important performance gains in difficult cases where there are few sequences or many paralogs per species, while IPA is already very good in other cases. Therefore, with computational cost in mind, we recommend the use of GA primarily in such difficult cases.

## Conclusion

In this work, we have shown that the search for interacting paralogs between two protein families strongly benefits from combining two different sources of information, namely phylogeny (via sequence similarity) and residue coevolution, assessed either using DCA or mutual information. This is interesting because these two sources of information are very rarely explicitly combined in computational analyses of protein sequence family. Most phylogenetic analyses work under the assumption of independent-site evolution, i.e. they disregard coevolution between residues. Conversely, most coevolutionary studies effectively treat sequences as close to independent, and employ corrections to reduce the impact of phylogenetic correlations that could obscure functional coevolution.

While unifying these two signals in a single framework remains a hard problem, they have been shown to combine constructively in the inference of protein partners among paralogs by coevolution methods [37], raising the possibility that explicitly combining them may further increase performance. Here, we combine these two signals in a technically straightforward way. By using sequence-similarity networks as a proxy of phylogeny, we can formulate the paralog-pairing problem as a graph-alignment problem, which allows us to identify a subset of high-quality pairings. This partial but robust co-MSA can be used to inform coevolutionary modeling. Indeed, starting from a well-matched seed co-MSA is strongly beneficial to DCA-based paralog pairing.

Indeed, we find that our two-step strategy (GA-IPA), combining phylogenetic and coevolutionary information, leads to pairings of higher accuracy. Moreover, it is substantially more robust. In particular, it performs well even in situations where coevolution-based paralog pairing alone performs poorly, including shallow MSAs and families with high paralog multiplicities.

In some cases, more information is available, such as a few experimentally known interacting pairs, or genetic co-localization of the genes coding for interacting proteins (e.g. in bacterial operons; this information was used to construct our ground truth). It is easy to implement this kind of information into the graph alignment algorithm by modifying the parameters $c_{mn}$ in Eq 1. Here, this parameter is only used to penalize inter-species pairings, but negative values could be used to favor or even impose pairings of specific sequences for which additional knowledge is available. This can be used to "nucleate" a graph alignment, and could make the robust GA-generated pairing larger and more accurate, thus further improving the performance of GA-IPA.

Here, for simplicity, we considered the simple case where each species comprises the same number of paralogs from each of the two families. However, in natural data, species include different number of paralogs from different families. In [35], this was addressed by using an injective pairing from the family with fewer paralogs into the family with more paralogs. Specifically, during the iterative procedure, in a species with more proteins A than proteins B, each protein B is matched one-to-one to a protein A so that the pairing score is maximized,

and one discards the remaining proteins A. This only requires a minor change in our algorithm and could be employed in GA-IPA. A broader challenge, which should be investigated in future work, is to account for the possibility of partially promiscuous interactions: proteins from family A may interact with more than one protein from family B (but not all), and vice versa. This kind of promiscuity is frequently found in eukaryotic signaling systems.

## Materials and methods

### Definitions and notations

We start from two MSAs **A** and **B** for two interacting protein families A and B (see Fig 1A). **A** contains $M$ protein sequences $\underline{a}^m = (a_1^m, \ldots, a_{L_A}^m)$ indexed by $m = 1, \ldots, M$, and of aligned length $L_A$. These sequences belong to $N < M$ distinct species, i.e., there are on average $\langle m_p \rangle = M/N > 1$ paralogs per species, and the number of paralogs can vary across species, cf. Fig A in S1 Text. For simplicity, we assume that **B** contains the same number $M$ of sequences $\underline{b}^m = (b_1^m, \ldots, b_{L_B}^m)$, $m = 1, \ldots, M$, but $L_B$ can differ from $L_A$. We further assume that each species has the same number of paralogs of family A and of family B (see Discussion above).

We aim at constructing a bijective matching $\pi$: $\{1, \ldots, M\} \rightarrow \{1, \ldots, M\}$ called *paralog pairing*, which assigns to each sequence $\underline{a}^m$ one putative interaction partner $\underline{b}^{\pi(m)}$. We only consider intra-species PPI, which implies that for all $m = 1, \ldots, M$, the indices $m$ and $\pi(m)$ belong to the same species.

### Data sets

We consider as our primary benchmark a data set composed of 23,632 pairs of natural sequences of interacting histidine kinases (HK) and response regulators (RR) from the P2CS database [42, 43], as previously described in [34, 38]. In this data set, interacting partners are determined using proximity in the genome, derived from the annotations of the P2CS database. This allows us to assess partner inference performance in this natural data sets as well as in derived ones. Discarding the 208 pairs from species with only one such pair for which pairing is trivial yields a dataset of 23,424 HK-RR pairs with 11.1 paralogs per species on average.

In our first benchmark, we focus on a smaller "standard dataset" extracted from this complete dataset, in view of computational time constraints. The standard dataset was constructed by picking species randomly. It comprises 5064 pairs from 459 species comprising at least two HK-RR pairs, with an average number of pairs per species $\langle m_p \rangle = 11.03$ [34]. To assess the impact of the number of HK-RR pairs per species on the success of GA-IPA, we constructed another dataset where the species with the highest numbers of pairs (25 to 41 in practice) were picked, yielding a dataset with $M = 5052$ pairs and $\langle m_p \rangle = 29.20$ [34]. In both datasets, the actual paralog numbers vary strongly from species to species, going from the imposed minimum of two paralogs up to a maximum of 42 paralogs, cf. the histograms in Fig A in S1 Text. Finally, to assess the impact of varying $M$ on the performance of inference by GA-IPA, we constructed smaller data sets by picking species randomly from the full data set [34].

We also consider a data set comprising 17,950 pairs of ABC transporter proteins homologous to the *Escherichia coli* MALG-MALK pair of maltose and maltodextrin transporters [12, 34] and extract a dataset of $\sim 5000$ pairs from it. Similarly, we consider the homologs of interacting *E. coli* enzymes XDHA-XDHC, and retain the full dataset of $\sim 2000$ pairs in this case. In these data sets, interacting partners are determined using proximity in the genome (as for HK-RR), following the approach from Ref. [12].

Since the approach presented here explicitly relies on phylogeny, it is interesting to also test it on proteins that share phylogeny without being interacting partners or having common

functional constraints. While it is difficult to be certain that two protein families do not have common functional constraints, we picked two pairs of families that are encoded in close proximity on prokaryotic genomes but do not have known physical interactions [44]. They are the *E. coli* protein pairs LOLC-MACA and ACRE-ENVR and their homologs [36]. (Note that ENVR has regulatory roles on ACRE expression [45].) The datasets we employed for these pairs include $\sim 2000$ homologous pairs.

## Constructing sequence-similarity networks

Let us present the construction of sequence-similarity networks for one MSA (note that these networks are constructed in an equivalent way for each of the two MSAs that we want to pair).

A first step for the construction of sequence-similarity networks is the choice of a distance (or dissimilarity) measure $d_{mn}$ for any pair $(m, n) \in \{1, \ldots, M\}^2$ of homologous sequences. Here we choose the Hamming distance, which simply counts the amino-acid mismatches between the two aligned sequences (see discussion in Results and discussion).

Equipped with this distance measure between aligned sequences, we construct for each MSA a sequence-similarity network $G = (V, E, w)$, defined as a weighted undirected graph with vertices (or nodes) $V = 1, \ldots, M$, edges $E$ and positive edge weights $w : E \to \mathbb{R}^+$. As mentioned above, we consider two types of similarity networks, where the edges are extracted using the following two distinct procedures:

- *kNN network*: In this network, each node is connected to its $k$ nearest neighbors, possibly including links inside one species (connecting closely related paralogs), or multiple links between species. The kNN network is *a priori* directed (i.e. if $n$ is a $k$-nearest neighbor of $m$, then $m$ is not necessarily a $k$-nearest neighbor of $n$). Here, we disregard the directionality of edges, and retain an edge if it is present at least in one direction. We further merge possible double edges resulting from reciprocal choices. Therefore, nodes have degrees superior or equal to $k$, cf. Fig B in S1 Text for an example degree distribution. The parameter $k$ is systematically varied in our analyses.

- *Orthology network*: As a simple operational definition of orthology we use reciprocal best hits. For each protein in the MSA, we select its closest neighbor in each of the other species. We include an edge between two sequences if and only if this selection is reciprocal. Note that this construction can lead to very high degrees, cf. Fig B in S1 Text. For instance, in a species having a single sequence, this sequence is connected to all $N - 1$ other species. Contrarily to kNN networks, orthology networks do not contain any links inside species, even when close paralogy is present.

To complete the construction of $G$, we define the edge weights as $w_{mn} = \exp(-d_{mn}^2 / D^2)$, where $d_{mn}$ is the distance between $m$ and $n$, while $D$ is the average distance (over all nodes) of the $k$th nearest neighbor in the case of the kNN network, and the average distance of the most distant ortholog in the case of the orthology network. The weight $w_{mn}$ gives stronger importance (i.e. larger weight) to edges between similar sequences.

## Aligning two sequence-similarity networks (GA)

We construct a network for each of the two families A and B, leading to two networks $G^A = (V, E^A, w^A)$ and $G^B = (V, E^B, w^B)$. Because of the similarities in the phylogenies of interacting proteins, if $(\underline{a}_m, \underline{b}_n)$ and $(\underline{a}_{m'}, \underline{b}_{n'})$ are two interacting protein pairs, and $\underline{a}_m$ and $\underline{a}_{m'}$ are close homologs (i.e. small $d_{mm'}^A$), then $\underline{b}_n$ and $\underline{b}_{n'}$ are also expected to be similar (i.e. small $d_{nn'}^B$).

Thus, the search for a paralog pairing $\pi$ can be mapped to a *graph-alignment* problem, where the vertices of the two graphs are paired to maximize the number of overlapping edges. To this end, we define a cost function

$$\mathcal{C}(\pi) = -\sum_{(m,n)\in E} w_{mn}^A w_{\pi(m)\pi(n)}^B + \sum_{m\in V} c_{m\pi(m)} \; , \tag{1}$$

which, for any given pairing $\pi: V \rightarrow V$, determines the negative of the total weight of all overlapping edges. The last term in the cost function is defined via

$$c_{mn} = \begin{cases} \infty & \text{for all } m \text{ and } n \text{ belonging to different species;} \\ 0 & \text{otherwise ;} \end{cases} \tag{2}$$

i.e. an infinite penalty is introduced for any pairing between proteins from different species. This term guarantees that only proteins from the same species are paired.

Finding the paralog pairing $\pi$ with minimal cost $\mathcal{C}(\pi)$ is highly non-trivial. Here we employ a heuristic stochastic local search algorithm based on *simulated annealing*.

## Approximately aligning two sequence-similarity networks by simulated annealing

Simulated annealing [46] is a heuristic optimization method aiming to find the state of a system that minimizes a cost function. In this approach, the amount of noise is gradually decreased via the temperature $T$, according to a predefined cooling protocol, until $T$ is close to zero (corresponding to very little noise). This procedure reduces the risk of getting stuck at local minima of the cost function. At each temperature, Markov Chain Monte Carlo (MCMC) updates are run until the system is in thermal equilibrium. Here we use the following cooling protocol, known as the exponential schedule [47]:

$$T(t) = T_0 \alpha^t, \tag{3}$$

where $T$ is the temperature at simulation step $t$, while $T_0$ is the initial temperature and $\alpha$ is a coefficient satisfying $0 < \alpha < 1$.

Pseudocode for the simulated annealing procedure is given below.

**Algorithm:** Simulated annealing

0. Initialize the matching $\pi$ by randomly pairing each sequence A with a sequence B from the same species.
   Initialize temperature and number of steps: $T = T_0$ and $t = 0$.

1. Propose MCMC updates at temperature $T$, until a fixed number are accepted.
   Each proposed update proceeds as follows:
   *Randomly select one pair of sequences from the full dataset.
   * Randomly select a second pair of sequences from the same species.
   * Propose to exchange their interaction partners, yielding a new matching $\pi'$.
   * Update the matching to $\pi'$ with probability min $\{1, \exp[(\mathcal{C}(\pi) - \mathcal{C}(\pi'))/T]\}$.

2. Multiply the temperature $T$ by a factor $\alpha$, and increase $t$ by one.

3. Repeat steps 1 and 2 until $T$ reaches a target small value.

## Coevolution-based iterative pairing algorithm (IPA)

The IPA, introduced in [34], starts from a seed co-MSA ("gold-standard set"), from which a pairwise maximum entropy model, also known as a DCA model [15, 32, 33] or as a Potts model, is inferred (see Fig 1C). Because this DCA model is built from a co-MSA, it is able to attribute a statistical energy score to any concatenated sequence composed of one sequence of family A and one of family B. This model is used to score all possible within-species A-B pairs that are not contained in the gold-standard set. These scores are used to perform a one-to-one matching of sequences within each species. The $N_{\text{increment}}$ pairs with the top confidence score (based on an energy gap [34]) among these proposed pairs are then added to the gold-standard set to form an extended co-MSA. This extended co-MSA is then employed to infer a new DCA model, which is in turn used to re-score all pairs of sequences not belonging to the gold-standard set. The procedure is iterated, adding the $nN_{\text{increment}}$ to the gold-standard set at iteration $n$, until the co-MSA is complete. This procedure heuristically constructs a pairing having high inter-protein DCA coevolutionary signal.

The IPA can also be run without a seed co-MSA. In this case, a random pairing within each species is used to infer the first DCA model. Contrarily to the case with a seed co-MSA where the seed co-MSA is not scored and left untouched, all pairs are always scored and re-paired at each iteration in this case.

In [38], a variant of the DCA-based IPA, based on pointwise mutual information, the MI-IPA, was introduced. It relies on the same iterative procedure, but uses scores based on mutual information instead of inferring DCA models. The MI-IPA reaches performances slightly higher the DCA-IPA on natural datasets, and is more robust to smaller datasets [38].

Matlab implementations of the DCA-based IPA and the MI-based IPA on our standard HK-RR dataset are freely available at https://doi.org/10.5281/zenodo.1421861 and https://doi.org/10.5281/zenodo.1421781, respectively. Julia implementations of the GA-IPA (as well as of the IPA), both DCA-based and MI-based are freely available at https://doi.org/10.5281/zenodo.7731108.

## Comparison with MirrorTree-based methods

Several methods based on sequence similarity (and thus phylogeny) have been developed to predict protein-protein interactions. A prominent family of such methods, known as MirrorTree [16–18], assesses how similar the pairwise distance matrices between sequences are across two protein families to determine whether they interact. MirrorTree approaches use different ways to measure similarity matrices, the basic methodology uses the neighbor-joining algorithm implemented in ClustalW [48] for generating a phylogenetic tree for each protein family that is then used to compute pairwise distance matrices between all orthologs by summing the lengths of the branches separating the corresponding orthologs [16, 17, 19–25]. Distance measures were corrected for multiple hits allowing that distances can be more than 1 via multiple substitution per site [23] by using protdist from the phylip [49] package instead of ClustalW. Also, [50] incorporates information on the overall evolutionary histories of the species to correct distances between orthologues due to speciation events. While these methods often restrict to orthologs, some variants tackle the paralog pairing problem [19–28]. In several of these methods [19–21, 23, 24], the order of the sequences in one of the two families is modified to make the two distance matrices more similar. Similarity can be assessed e.g. via the Pearson correlation between the entries of the two distance matrices [21]. In this framework, consider the matrix $\underline{d}^A$ (resp. $\underline{d}^B$) with elements $d_{mn}^A$ (resp. $d_{mn}^B$) denoting the distance between sequences $m$ and $n$ of family A (resp. family B). The similarity score between $\underline{d}^A$ and $\underline{d}^B$ is then

the Pearson correlation between the ordered list of all elements of $\underline{d}^A$ on the one hand and the ordered list of all elements of $\underline{d}^B$ on the other hand [21]. We consider this MirrorTree-based approach to the paralog matching problem and compare it to our graph alignment (GA) method in Table A in S1 Text. Optimization is performed via a Monte Carlo algorithm. One can exploit the stochasticity of this optimization by considering pairs that are robustly predicted over many replicates, exactly as for GA, see Table B in S1 Text.

Besides, as we proposed in [36], one can also use a variant of MirrorTree to address the paralog pairing problem in the IPA spirit. For this, we need to attribute a score to each possible within-species A-B pair, using a training co-MSA, which is the one of the gold-standard set at the first IPA step, or the extended co-MSA enriched with top predictions at the next ones (see above). In the MirrorTree approach, we use a Pearson correlation between Hamming distances observed in the two families to define this score. Specifically, let $\{\underline{a}_1^{\mathrm{tr}}\underline{b}_1^{\mathrm{tr}}, \ldots, \underline{a}_P^{\mathrm{tr}}\underline{b}_P^{\mathrm{tr}}\}$ be the training co-MSA, which contains $P$ pairs A-B. For each chain $\underline{a}_i$ of the testing set, we compute the vector $\underline{d}_i^A = (d(\underline{a}_i, \underline{a}_1^{\mathrm{tr}}), \ldots, d(\underline{a}_i, \underline{a}_P^{\mathrm{tr}}))$ of Hamming distances between $\underline{a}_i$ and each chain $\underline{a}_m^{\mathrm{tr}}$ of the training set. We also compute an analogous vector $\underline{d}_j^B$ for each chain $\underline{b}_j$ of the testing set. Next, we define the pairing score of the possible pair A-B composed by $\underline{a}_i$ and $\underline{b}_j$ as the Pearson correlation between $\underline{d}_i^A$ and $\underline{d}_j^B$. This score can be used for predicting partnerships in the IPA exactly as the DCA- and MI-based scores. In particular, to work fully in the MirrorTree spirit while using similar algorithms as in the rest of our work, we can start the MirrorTree-based IPA from a robust set of pairs predicted by MirrorTree (see paragraph above), yielding the results in Table C in S1 Text.

## Supporting information

**S1 Text. All supporting material is collected in a single supplementary text file.**
(PDF)

## Author Contributions

**Conceptualization:** Anne-Florence Bitbol, Martin Weigt.

**Data curation:** Carlos A. Gandarilla-Pérez, Anne-Florence Bitbol.

**Formal analysis:** Carlos A. Gandarilla-Pérez.

**Funding acquisition:** Anne-Florence Bitbol, Martin Weigt.

**Investigation:** Carlos A. Gandarilla-Pérez, Sergio Pinilla, Anne-Florence Bitbol.

**Methodology:** Carlos A. Gandarilla-Pérez, Sergio Pinilla, Anne-Florence Bitbol, Martin Weigt.

**Software:** Carlos A. Gandarilla-Pérez, Anne-Florence Bitbol.

**Supervision:** Anne-Florence Bitbol, Martin Weigt.

**Writing – original draft:** Carlos A. Gandarilla-Pérez, Anne-Florence Bitbol, Martin Weigt.

**Writing – review & editing:** Anne-Florence Bitbol, Martin Weigt.

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
