## [Decision Letter · Decision Letter 0]

10 Oct 2022

Dear Weigt,

Thank you very much for submitting your manuscript "Combining phylogeny and coevolution improves the inference of interaction partners among paralogous proteins" for consideration at PLOS Computational Biology.

As with all papers reviewed by the journal, your manuscript was reviewed by members of the editorial board and by several independent reviewers. In light of the reviews (below this email), we would like to invite the resubmission of a significantly-revised version that takes into account the reviewers' comments, and especially

a comparison with existing methods clarifying the potential advances of the method you propose.

We cannot make any decision about publication until we have seen the revised manuscript and your response to the reviewers' comments. Your revised manuscript is also likely to be sent to reviewers for further evaluation.

Sincerely,

Andrea Ciliberto

Academic Editor

PLOS Computational Biology

Arne Elofsson

Section Editor

PLOS Computational Biology

Reviewer's Responses to Questions

**Comments to the Authors:**

Reviewer #1: In this work, a sequence-based method for predicting the pairings between two paralogous families of interacting proteins is presented. The method combines two previous approaches: one based on maximizing phylogeny-matching via a heuristic graph alignment procedure (GA), and another based on maximizing the inter-family residue co-evolutionary signals yielded by the paired alignments (IPA). These are combined by taking the more stable pairs generated by GA as a “seed” for IPA. The approach is tested in 4 pairs of interacting families, including the well-studied bacterial two-component system, for which the real pairing is inferred from genome co-localization. The combined approach (GA-IPA) is shown to present a higher performance compared with the original approaches for the cases with a high paralog multiplicity (many paralogs per species) and small multiple sequence alignments (low number of sequences in the families).

Predicting pairings between paralog families is important for better understanding signaling pathways and how these evolve avoiding crosstalk. Sequence-based approaches for this task are benefited from the continuous stream of new genome sequences. While it lacks conceptual novelty (just combines two existing approaches), the methodology presented here is potentially interesting as it produces better results in some particular cases. The paper is clearly written.

Major comments

My interpretation of the results is that GA-IPA only outperforms IPA for the cases with a high paralog multiplicity (many paralogs per species) and “small” multiple sequence alignments. The second is fine but the first… how realistic is that scenario? The authors “simulate” an example with 29 paralogs per specie and many species. How many of such families are out there? To demonstrate the utility of their method in the “real world” it would be desirable to quantify this (e.g. using databases of protein families/paralogs) or, at least, discuss this point in detail.

There are many other approaches for the sequence based prediction of pairings between paralogs (e.g. PMIDs 14594708, 18277381, 18215279, 22399677, 16634043, 19696150, …). The authors should include a better overview of the field in their introduction.

Related with that, it would be desirable to compare the approach presented here with some of these (unrelated) previous approaches. Comparing with the two that are combined here (GA and IPA) is fine for evaluating the added value of the combination, but for putting this new methodology in the context of the current landscape of approaches, a comparison with “unrelated” approaches would be desirable.

Figure 1 is a little bit misleading. In real families orthologs are more self-similar than paralogs. The clades in the trees should correspond to orthologous groups (i.e. mixed colors/species).

Minor

Requiring two families with the same number of members in each species is really a drawback and, related to the comment above on the paralog multiplicity, I do not know how many cases of these are in the real world (specially in eukarya). The authors acknowledge this and say that their method could be modified to work with different number of members. While doing that is beyond the scope of this particular work, at least giving some receipts on how to apply the current method in those cases should be given: i.e. what can I do if I want to apply this method to my families with different #proteins? Which ones could I discard?....

It is not true that most co-evolutionary studies “disregard phylogeny” (Conclusions). Indeed, the new-wave of method for detecting residue co-evolution are characterized, among other things, for correcting the phylogenetic signal in one way or another (in most cases not explicitly but implicitly).

Reviewer #2: The paper by Gandarilla-Perez et al. presents an interesting and novel computational framework to predict protein-protein interactions from Multiple Sequence Alignments (MSA). The methods proposed aims at identifying pairs of interacting proteins exploiting the inter-protein evolutionary correlations observed among their residues. While in previous approaches, presented by some of the authors, such evolutionary correlations were quantified using residue-residue correlations only (by means of Mutual Information (MI) and Direct Coupling Analysis (DCA)), here the aim is to integrate this information with phylogenetic correlations, which were not expected to be captured by MI and DCA.

Phylogenetic correlations are then used to produce partial co-alignments of interacting proteins (co-MSAs), obtained from similarity graph alignment, for which authors developed a stochastic algorithm based on simulated annealing to solve the graph-alignment (GA) problem. The co-MSA is the used as an input to the iterative pairing algorithm (IPA) based on residue coevolution as detected by DCA (or by MI).

I find the algorithmic approach presented in the paper to be technically sound and well grounded on protein evolution concepts.

As for the results presented, the combination of GA+IPA clearly display improved performances of interaction partner inference when the average number of potential partners is large and when the total data set size is modest (see Fig5 and related results). However, in (arguably) more "standard" situations the gain of this new approach vs already existing (and published) approaches is very marginal (Fig.4-Fig.S4 and related results). For this reason, I think authors should address the following major point:

- what is the computational cost of the GA algorithm? How does it scale w.r.t. key parameters (such as the total number of sequences or the average multiplicities)?. Here I'd be only interested with a computational analysis (say execution time vs N or similar).

In other words, I think it would be important for the new approach to be actually deployed in a systematic way, to know whether the computational cost is worthy the (possible) little gain in the accuracy of the inference.

**Have the authors made all data and (if applicable) computational code underlying the findings in their manuscript fully available?**

Reviewer #1: Yes

Reviewer #2: Yes

PLOS authors have the option to publish the peer review history of their article (what does this mean?). If published, this will include your full peer review and any attached files.

Reviewer #1: No

Reviewer #2: No
---

## [Decision Letter · Decision Letter 1]

8 Feb 2023

Dear Weigt,

Thank you very much for submitting your manuscript "Combining phylogeny and coevolution improves the inference of interaction partners among paralogous proteins" for consideration at PLOS Computational Biology. As with all papers reviewed by the journal, your manuscript was reviewed by members of the editorial board and by several independent reviewers. The reviewers appreciated the attention to an important topic. Based on the reviews, we are likely to accept this manuscript for publication, providing that you modify the manuscript according to the review recommendations.

Sincerely,

Arne Elofsson

Section Editor

PLOS Computational Biology

Arne Elofsson

Section Editor

PLOS Computational Biology

Reviewer's Responses to Questions

**Comments to the Authors:**

Reviewer #1: The authors addressed all may concerns and performed new experiments. I think the paper is clearer now and can be published.

My only (minor) concern is that, as stated now in the "Comparison with MirrorTree-based methods" it looks like these mirrirtree-based approaches for paralog mapping use Hamming distances for quantifying protein similarities. This is not true and they use other measures (average similarity, substitutions/site, ...) I understand the authors' reason to implement mirrortree approaches with Hamming distances in the context of this work as those are the ones used by their method, and, as commented in other part of the manuscript, I would not expect major differences using other measures (so I do not think an implementation of the approaches with their "original" distances is required). But at least the point should be clarified and an explanation why mt approaches were implemented with hamming distances for this particular work added.

Reviewer #2: The authors have addressed all my remarks in a thorough and clear manner. Overall, the manuscript is very well written and contains interesting results which are well supported by the presented evidence.

I recommend the manuscript for publication.

**Have the authors made all data and (if applicable) computational code underlying the findings in their manuscript fully available?**

Reviewer #1: Yes

Reviewer #2: Yes

PLOS authors have the option to publish the peer review history of their article (what does this mean?). If published, this will include your full peer review and any attached files.

Reviewer #1: No

Reviewer #2: No

Figure Files:

Data Requirements:

Reproducibility:

References:

---

## [Editor Report · Decision Letter 2]

8 Mar 2023

Dear Weigt,

We are pleased to inform you that your manuscript 'Combining phylogeny and coevolution improves the inference of interaction partners among paralogous proteins' has been provisionally accepted for publication in PLOS Computational Biology.

Best regards,

Andrea Ciliberto

Academic Editor

PLOS Computational Biology

Arne Elofsson

Section Editor

PLOS Computational Biology

---

## [Editor Report · Acceptance letter]

24 Mar 2023

PCOMPBIOL-D-22-01270R2 

Combining phylogeny and coevolution improves the inference of interaction partners among paralogous proteins

Dear Dr Weigt,

I am pleased to inform you that your manuscript has been formally accepted for publication in PLOS Computational Biology. Your manuscript is now with our production department and you will be notified of the publication date in due course.

With kind regards,

Zsofia Freund
